# Postoperative surveillance and long-term outcome after endovascular aortic aneurysm repair in the Netherlands: study protocol for the retrospective ODYSSEUS study

Anna Catharina Maria Geraedts ![ORCID] , Sylvana de Mik, Dirk Ubbink, Mark Koelemay, Ron Balm, on behalf of the ODYSSEUS study group

Surgery, Amsterdam University Medical Centres, Amsterdam, Noord-Holland, The Netherlands

**Correspondence to**
Professor Ron Balm;
r.balm@amsterdamumc.nl

## ABSTRACT

**Introduction** Strict imaging surveillance protocols to detect complications following endovascular aneurysm repair (EVAR) are common practice. However, controversy exists as to whether all EVAR patients need intense surveillance. The 2019 European Society for Vascular Surgery guidelines for management of abdominal aortic aneurysm (AAA) suggest that patients may be considered for limited follow-up with imaging if classified as 'low risk' for complications based on their initial postoperative imaging. The current study aims to investigate the intervention-free survival and overall survival stratified for patients with and without yearly imaging surveillance.

**Methods and analysis** The Observing a Decade of Yearly Standardised Surveillance in EVAR patients with Ultrasound or CT Scan study comprises a national multicentre retrospective cohort study in 17 medical centres. Consecutive patients with an asymptomatic or symptomatic infrarenal AAA who underwent EVAR between January 2007 and January 2012 will be included in this study with follow-up until December 2018. Clinical variables and all follow-up information will be retrieved in extensive data collection from the patient's medical records. In addition, an e-survey was sent to vascular surgeons at the 17 participating centres to gauge their opinions regarding the possibility of safely reducing the frequency of imaging surveillance. Primary endpoints are intervention after EVAR and aneurysm-related mortality. The initial estimated sample size is 1997 patients.

**Ethics and dissemination** The study has been approved by the Medical Ethics Review Committee of the Amsterdam UMC, location Academic Medical Centre, Amsterdam, the Netherlands. Study findings will be disseminated via presentations at conferences and publications in peer-reviewed journal.

**Trial registration number** The Netherlands Trial Registry, NL6953 (old: NTR28773).

## INTRODUCTION

Endovascular aneurysm repair (EVAR) of the abdominal aorta has become the primary treatment of patients with an abdominal

## Strengths and limitations of this study

► The main strength of this study is that it can accumulate data from large number of patients with long-term follow-up up to 11 years and that it captures all surveillance visits, long-term outcomes and mortality post-endovascular aneurysm repair.
► The sample size will be large enough to enable survival and regression analyses in sub-groups of patients.
► The main limitation of the study is due to the nature of retrospective data, it allows only the collection of data that was documented in the patient medical records.

aortic aneurysm (AAA).[1] Both the Society for Vascular Surgery (SVS) International Guidelines and the instructions for use (IFU) of endograft manufacturers recommend yearly imaging surveillance for all patients after EVAR.[2] However, if the patient is classified as 'low risk' for complications based on initial post-operative imaging, the 2019 European Society for Vascular Surgery (ESVS) guidelines recommend delaying imaging until 5 years after repair.[3] This movement towards reducing the imaging frequency will benefit patients, medical centres and healthcare costs.

Imaging surveillance by CT angiography (CTA) may increase the attributable lifetime cancer risk of patients, as well as putting them at risk of developing nephropathy due to contrast exposure. If yearly CTA is replaced by duplex ultrasonography (DUS) patients still experience the burden of additional hospital visits. Moreover, compliance with yearly imaging is suboptimal and non-adherence to yearly imaging does not appear to be associated with poorer outcomes.[4 5]

It has been questioned whether yearly imaging is necessary for all EVAR patients, and if a specific group of patients can be identified for which surveillance intervals can safely be extended, as is suggested by the new guideline.[3] For these reasons, in the Netherlands the Observing a Decade of Yearly Standardised Surveillance in EVAR patients with Ultrasound or CT Scan (ODYSSEUS) study has been designed. In this study of approximately 2000 patients with 6–11 years of follow-up, we aim to determine when, and in which patients, it is safe to deviate from the current annual surveillance protocols.

### Background and relevant literature and data

Before initiating the ODYSSEUS study, we conducted a survey among Dutch vascular surgeons to find out if they support the possibility of reducing the frequency of imaging surveillance. In this survey, vascular surgeons reported the main reasons patients did not comply with follow-up visits, that is, they had forgotten the appointment or were prevented by force majeure. Most physicians estimated that less than 10% of their patients had missed one or more follow-up visits post-EVAR. This might be an overestimation of the true adherence to follow-up visits, as these observations are in contrast with a study reporting that only 43% of patients had complete surveillance.[4]

We also asked participating vascular surgeons to upload their standard post-EVAR protocol to investigate if there were differences between centres in the Netherlands. In all centres, imaging took place within the first 3 months after surgery, mostly by CTA. Most centres comply with their own post-EVAR surveillance protocols which have many commonalities with the SVS and ESVS guidelines. Only one centre uses precisely the same post-EVAR surveillance protocol as recommended by the SVS guidelines. Another centre had already reduced follow-up imaging to once every 5 years, using either CTA or DUS as is stated in the new ESVS guidelines.[3] While vascular surgeons still seem to adhere to their hospital-specific protocol, they do support the need for reducing follow-up by selecting a group of patients for which yearly follow-up can safely be omitted. However, some surgeons indicated that more evidence is needed than is available in the current literature.[6 7]

In studies that have investigated the indications for post-EVAR intervention, it is stated that 61%–98% of interventions were necessary because of symptoms and not because of findings at surveillance imaging. This suggests that post-EVAR surveillance protocols provide no benefit to a large group of patients, as complications occur in between surveillance visits.[8 9] Imaging surveillance may even lead to unnecessary interventions and it does not appear to be associated with improved survival.[9 10] We hypothesise that the requirement for routine imaging for patients at low risk can be reduced. However, novel endovascular devices still require more intensive surveillance as the short- and long-term results of those devices remain undetermined.

## STUDY OBJECTIVES

The objective of this study is to evaluate whether imaging surveillance frequency might have been safely reduced in a selected group of EVAR patients, for example, in patients with an asymptomatic or symptomatic infrarenal AAA who underwent EVAR and who had no abnormalities on the 3-month postoperative CTA. The clinical course of a large cohort of patients will be evaluated with follow-up ranging between 6 and 11 years. Baseline patient characteristics, aortic anatomy and details of the operation will be derived from the patient's medical record. The first milestone during follow-up is the first postoperative CTA. This scan either shows complications such as endoleaks, malposition or migration of the graft, or the absence thereof. All follow-up visits, imaging studies, as well as all interventions after EVAR and outcomes will be registered. Our hypothesis is that patients with less follow-up will have better outcomes regarding the number of interventions and aneurysm-related mortality compared to patients with annual follow-up. Regarding the intervention rates, it is expected that adherence to imaging surveillance may detect more abnormalities triggering re-interventions which in itself may cause additional complications and perhaps even a decrease in survival rates. We hypothesise that the need for routine imaging for patients with no abnormalities at their initial CTA can be decreased.

## METHODS AND ANALYSIS

The study protocol has been designed according to the Standard Protocol Items: Recommendations for Interventional Trials statement and the CONsolidated Standards of Reporting Trials.[11 12]

### General study design

A multicentre retrospective cohort study in 17 medical centres in the Netherlands. Data will be collected from the medical records of all consecutive patients with AAA who underwent EVAR between January 2007 and January 2012. This selection provides a theoretical length of follow-up of 6–11 years on December 2018. Patients will be divided into three groups: (A) patients without abnormalities at their first postoperative CTA with yearly imaging surveillance, (B) patients without abnormalities at their first postoperative CTA without yearly imaging surveillance and (C) patients with abnormalities at their first postoperative CTA (figure 1). This retrospective design has the advantage of collecting long term follow-up data. The Dutch Dream trial found that the number of interventions starts to rise 4 years after EVAR and the long-term results of the EVAR-1 trial show that EVAR has an early survival benefit but inferior late survival compared to open surgical repair.[13 14] This is in contrast to the recently published long-term results of the OVER trial in which no difference was observed between EVAR and OSR in the primary outcome of all-cause mortality.[15] Hence, a prospective study would take approximately 8 to 10 years to gather enough patients with adequate follow-up.

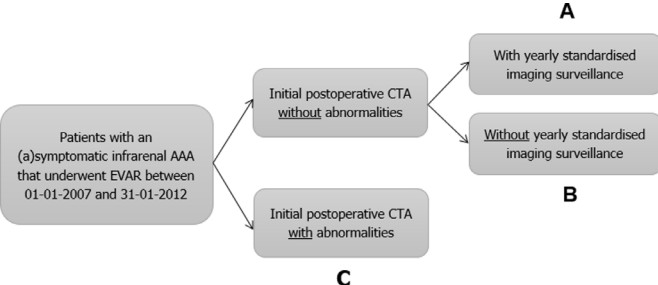

**Figure 1** Patient subgroups. AAA, abdominal aortic aneurysm; CTA, CT angiography; EVAR, endovascular aneurysm repair.

The standard of care is defined by the current guidelines and IFU. The usual follow-up schedule in the IFU is: CTA and abdominal X-ray at 30 days, 6 and 12 months and yearly thereafter. The 2019 ESVS guidelines recommend a CTA 30 days after EVAR. If there is adequate seal and no endoleak, patients are classified as low risk and CTA follow-up may take place 5 years later. If there is an inadequate seal and endoleak type I/III patients could be either evaluated for re-intervention or if sac shrinkage occurs yearly DUS is recommended. In the 2018 SVS guideline, CTA at 1 and 12 months is recommended and if neither endoleak nor sac enlargement is documented, DUS is suggested for annual postoperative surveillance. In our study design, the definition of compliance is undergoing imaging surveillance every 16 months since patients in most centres will be rescheduled if they missed their annual follow-up visit. Device-specific complications after EVAR will also be examined.

## Study population
Patients eligible for this retrospective study are all adults who underwent elective EVAR for asymptomatic or symptomatic infrarenal AAA between January 2007 and January 2012. Table 1 gives a more detailed overview of the inclusion and exclusion criteria.

## Patient and public involvement
No patients were involved in the research design and conception of this research study.

| Table 1 | Retrospective cohort |
|---|---|
| **Inclusion criteria** | **Exclusion criteria** |
| Age ≥18 years | Connective tissue disease |
| Patient with an (a)symptomatic infrarenal abdominal aortic aneurysm EVAR between January 2007 and January 2012 | Patients who objected to their retrospective data being used |
| Patients with an initial postoperative CTA within 90 days after EVAR | |

CTA, computed tomography angiography; EVAR, endovascular aneurysm repair.

## Date range of the study
Data will be extracted from patient medical records retrospectively and entered into a database with data validation from December 2018 until June 2020. At first, two researchers will extract data together to standardise data extraction. Next, to further improve the validity of the data two researchers will independently extract data and enter it into the secured data base. Disagreements will be noted and resolved by discussion and if necessary by asking another co-author to act as an arbiter.

## Subject selection
A retrospective cohort study of consecutive patients treated at 17 vascular centres is to be performed. All patients are eligible and the opt-out procedure will be used to allow patients to object to participation within 4 weeks, which is in accordance with the Dutch Code of Civil Procedure. The Medical Ethics Review Committee of the Amsterdam UMC, location Academic Medical Centre, Amsterdam, has confirmed that the Medical Research Involving Human Subjects Act (WMO) does not apply to our study. This study is conducted according to the General Data Protection Regulation (AVG 2016) and the Medical Treatment Agreement Act (WGBO).

## Data sources
Paper or electronic medical records are used in order to identify participants who match study-defined criteria.

## Primary and secondary endpoints
### Main study endpoint
► The number of patients with an intervention and aneurysm-related mortality classified for patients with and without yearly imaging surveillance.

### Secondary study endpoints
► Date, type, indication and outcome of all postoperative imaging during follow-up.
► Type I, type II, type II and type IV endoleak, graft or outflow (iliac) occlusion, endograft infection detected by postoperative imaging, if present.
► Date and type of intervention during follow-up, if present.
► Date of aneurysm rupture during follow-up, if present.
► Date of death during follow-up, if present.
► Costs of all EVAR-related imaging and outpatient clinic visits.

## Study procedures
The primary outcomes of this study are interventions and aneurysm-related mortality for patients who had a normal initial postoperative CTA and who do adhere to our definition of yearly imaging surveillance over a 6-year to 11-year follow-up period, compared to those who do not adhere to our definition.

Interventions are EVAR-related interventions defined by the SVS reporting standards as postoperative adjunctive manoeuvres.[16 17] Interventions for wound complications

at the access site are not included, since these are detectable without the use of imaging.

Date of death during follow-up, if applicable, will be obtained from patient medical records and verified by the Dutch municipal personal records database (GBA).

Details of all surveillance imaging are obtained from patient medical records and radiology reports. The time between imaging appointments is calculated to determine whether patients adhere to this studies definition of with yearly imaging surveillance, that is, within every 16 months.

Date, type, indication and outcome of all postoperative imaging during follow-up are obtained from patient medical records, specifically imaging order forms and radiology reports. A normal initial postoperative CT scan is defined as a CT scan which shows no endoleaks, endograft migration (>10 mm), kinking or obstruction. All imaging outcomes are based on the report compiled by radiologists. These reports will not be re-evaluated by an independent radiologist, since we want to base our outcomes on real life data.

Secondary outcomes are all-cause mortality, type I, type II, type III and type IV endoleak, graft or outflow (iliac) occlusion, aneurysm rupture and endograft infection. This is also obtained from patient medical records, specifically radiology reports and Dutch municipal personal records database (GBA).

Date of aneurysm rupture is obtained from patient medical records, specifically operative reports, radiology reports and progress notes.

Costs of all EVAR-related imaging and outpatient clinic visits will be calculated per patient. Cost is defined as volume times price. Prices from the 'Cost manual of the Dutch Health Care Institute" will be used. Costs for the patients will also be included. The quality-adjusted life years, a generic measure of disease burden including both the quality and quantity of life, cannot be calculated with this retrospective design.

## Sample size and power

Sample size calculation for this study is based on an expected difference of 7% in the proportions of patients not requiring interventions after 7 years between patients undergoing yearly standardised imaging surveillance (75% intervention-free rate[18]) versus those s not undergoing standardised imaging surveillance (82% intervention-free rate[19]). To detect this difference with 90% power and a 0.05 significance level, 719 patients per group are required and 1438 in total. To correct for the fact that the first CTA of approximately 20% of patients is abnormal, 1798 patients (1438/0.8) are needed.[20 21] In addition, we expect incomplete data in 10% of the patients which results in a total number of 1997 patients (1798/0.9). With this sample size, we can also detect a 3% difference in aneurysm-related mortality with statistical significance. We chose a one-sided significance level (non-inferiority) of 0.05 and

for standard proportion a 95% non-aneurysm-related mortality and thus 5% freedom from aneurysm-related mortality after 7 years: 0.95. For equivalence limit difference, we chose an acceptable difference between groups of 3%, in which if differences in aneurysm-related mortality equals 3%, they are considered non-inferior. Test-expected proportion is then equal to the standard proportion 0.97. Thus, the expected difference is 0, calculated with a power of 80%. This results in 653 patients per group and 1306 patients in total. Since the first CTA of approximately 20% of patients is abnormal, 1632 patients (1306/0.8) are needed. In addition, we expect incomplete data in 10% of the patients which results in a minimum of, 1813 patients (1632/0.9) are needed in total.

## Statistical methods

Differences in baseline characteristics between patients undergoing or not undergoing yearly standardised imaging surveillance by either CTA or DUS, will be analysed using the $\chi^2$ or Fisher's exact test for categorical variables and the student's t-test or Mann-Whitney test for continuous variables, if appropriate.

The primary endpoints, that is, survival and freedom from intervention will be estimated by Kaplan-Meier survival analysis and differences between groups will be assessed with the log-rank test.

Secondary endpoints such as freedom from aneurysm rupture between patients with and without yearly standardised imaging surveillance will be estimated by Kaplan-Meier survival analysis, and differences between these groups will be calculated with the log-rank test.

Multivariable cox regression analysis will be used to determine survival and the freedom of intervention corrected for age, gender, AAA diameter, ASA classification, neck length, neck angulation and type of endograft. The association between postoperative intervention and the following covariates will be investigated with the multivariate Cox-regression analysis:

► Age.
► Gender.
► AAA diameter.
► ASA classification.
► Neck length (>15 mm).
► Neck angulation (>60°).
► Type of endograft.
► Initial postoperative CTA.

All statistical analyses will be done with SPSS software (IBM, V.25). The level for statistical significance is set at a p value <0.05. The proportion of missing data will be displayed. The missing values will be imputed by multiple imputation techniques if this does not exceed 10%–15% and conduct a sensitivity analysis to investigate the effect of the missing data on the results of the analysis. If missing data on outcome variables exceeds 15%, we plan to perform subgroup analysis**.**

## DISCUSSION

The goal of this study is to evaluate whether a reduction in follow-up visits and imaging and thus costs, in patients operated on for an asymptomatic or symptomatic AAA with EVAR is safe. We hypothesise that there will be less interventions and no difference in aneurysm-related mortality in patients with less intensive follow-up. With the results of this study, we aim to provide scientific evidence helping vascular surgeons decide whether less vigilant follow-up after EVAR may be considered for patients classified in the low risk group.

The strengths of the ODYSSEUS study are that it can accumulate the data of a large number of consecutive EVAR patients with a theoretical follow-up of 6 to 11 years, and that it captures all surveillance visits, long-term outcomes and mortality post-EVAR. Moreover, 17 medical centres throughout the Netherlands are participating in this study, including university and general hospitals, thereby reducing selection bias.

An e-survey has been sent to all vascular surgeons participating in the ODYSSEUS study. This shows that yearly imaging surveillance is upheld by most vascular surgeons in the Netherlands. In addition, most physicians agree that yearly imaging frequency can be safely reduced in a specific group of EVAR patients. As support for this reduction in frequency is evident in the Netherlands, the next step will be to study the groups for which it will be safe to deviate from the widely accepted surveillance protocols.

Our study is subject to limitations due to the nature of administrative data and its retrospective and observational design. As with all studies using administrative data, it allows only the collection of data that was documented in patient medical records. It is also possible that some patients may have transferred to alternative surveillance protocols in different medical centres without our knowledge. The study will assess results in 17 medical centres over 11 years, during which time improvements in endograft and in clinical practice has occurred. Attrition bias due to loss to follow-up represents a threat to the internal validity of our cohort study. The mentioned e-survey has only been sent to participating vascular surgeons, perhaps surgeons participating in the ODYSSEUS study strongly believe that imaging surveillance frequency can be reduced. This may have provided a biased view of post-EVAR follow-up in the Netherlands. However, most of the high-volume EVAR centres in the Netherlands have been included. Another limitation is that no information is retrieved from patients' medical records about when not to intervene and what the reason was for this decision.

In conclusion, with the ODYSSEUS study we aim to confirm the follow-up protocol of the recent ESVS guideline delaying imaging after 5 years if classified in the low risk group and therefore aim to investigate the intervention-free-survival and aneurysm-related mortality for patients with and without yearly imaging surveillance.

## ETHICS AND DISSEMINATION

Principles of good clinical practice will be respected. Study participation is voluntary. We aim to produce high-impact peer-reviewed publications of the results of the study and present our findings at national and international conferences. The members of the project group of this study will be involved in preparing manuscript drafts and abstract among any other publications arising from the study. The Netherlands Organization for Health Research and Development demands us to stay in close cooperation with the patients association ('Harteraad'). The results of this study will be shared with the members of the patients association via multiple modalities.

**Collaborators** on behalf of the ODYSSEUS study group: JW Elshof, BHP Elsman, JF Hamming, JA van Herwaarden, RHJ Kropman, MM Lensvelt, PP Poyck, GWH Schurink, AAEA de Smet, SM van Sterkenburg, C Ünlü, AC Vahl, HJM Verhagen, PWHE Vriens, JPPM de Vries, JJ Wever, W Wisselink, CJ Zeebregts.

**Contributors** Conception and design of study: ACMG, SdM, DU, MK, RB. Drafting the manuscript: ACMG, SdM. Revising the manuscript critically for important intellectual content: DU, MK, RB. Approval of the version of the manuscript to be published: ACMG, SdM, DU, MK, RB. Agreement to be accountable for all aspects of the work: ACMG, SdM, DU, MK, RB.

**Funding** This study is funded by the AMC Foundation and with financial support of the Netherlands Organization for Health Research and Development (ZonMw; grant 843004119). The duration of the study is from July 15, 2018 to July 15, 2021.

**Disclaimer** Neither the AMC Foundation nor ZonMw is involved in the study design, writing of the manuscript or the decision to submit the manuscript for publication.

**Competing interests** None declared.

**Patient consent for publication** Not required.

**Ethics approval** The Medical Ethics Review Committee of the Academic Medical Centre, Amsterdam, has reviewed and approved our study protocol version 1.6 dated 26 March 2018. The study is being conducted according to the principles of the Declaration of Helsinki in the current version of Fortaleza, Brazil (2013).

**Provenance and peer review** Not commissioned; externally peer reviewed.

**ORCID iD**
Anna Catharina Maria Geraedts http://orcid.org/0000-0002-6476-6027

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
