## [Reviewer comments · BMJ Open]

ARTICLE DETAILS

TITLE (PROVISIONAL)	Postoperative surveillance and long-term outcome after endovascular aortic aneurysm repair in the Netherlands: study protocol for the retrospective ODYSSEUS study.
AUTHORS	Geraedts, Anna; de Mik, Sylvana; Ubbink, Dirk; Koelemay, Mark; Balm, Ron

VERSION 1 - REVIEW

REVIEWER	Janet Powell Imperial College London, UK
REVIEW RETURNED	28-Aug-2019

GENERAL COMMENTS	Useful study, but may prove dated by time of publication. I have several comments which need to be addressed to improve the protocol (and hopefully the study too). 1 The relevant guideline checklist is STROBE not CONSORT2 The outcomes are not clearly listed, particularly the secondary outcomes. For instances costs are mentioned but never listed as a secondary outcome. I suggest giving a complete list of the prespecified outcomes before describing how the data are to be collected.3 The covariates to be included in the analyses are not listed and at least a partial list needs to be included in the manuscript.4 You need to consider the major limitation of this study, that it is retrospective and represents the imaging capabilities and expertise of >10 years ago (key issue of first post-operative CT scan). This limitation also is likely to lead to missing data, particularly for covariates, and you need to discuss how missing data are to be handled in the statistics section.5 How many persons will extract the data and enter it into the data base? There could be a need for at least partial double data extraction to improve the validity of the data.6 Do you need a protocol for extracting data from the radiology and/or duplex reports?
---

REVIEWER	Sam Tyagi University of Kentucky, USA
REVIEW RETURNED	13-Sep-2019

GENERAL COMMENTS	1. endpoints seem to focus on interventions and aneurysm related mortality. I would suggest categorizing and separating into thrombotic/occlusive/stenotic complications or interventions vs endoleak related vs infection related - there is wide variability in the mortality risk of these categories between and within these categories. If they are all lumped in together, it may confound some real findings. In order to do this, you will need to recalculate your statistical power and sample size. 2. You protocol lack mention of device specific complications. Some devices may be more prone to others to certain complications (ex. type 3 endoleak), and lumping them all together will confound data, and may mask a real finding. 3. Be aware that your follow up period may still be short. For example the endologix AFX graft with the old material (strata) had a 24% T3EL rate, but the average time to reintervention was 40+ months and up to 70 months.
---

REVIEWER	Frederic S. Resnic, MD MSc Lahey Hospital and Medical Center, Burlington MA, USA
REVIEW RETURNED	25-Oct-2019

GENERAL COMMENTS	The authors present the very well written protocol for the ODYSSEUS Study of the outcomes following EVAR with different strategies for post-procedural imaging surveillance. The issue being studied is relevant and important to the contemporary treatment of abdominal aortic aneurysms, and the approach proposed is both a pragmatic and efficient strategy for investigating the research question. I have only a few questions for the authors to consider in the effort to clarify some of the assumptions used in the sample size calculations presented in the protocol. 1) [Page 12, Line 45] More detailed justification or references for the selection of the baseline rate of anticipated intervention-free survival (primary endpoint) of 75% at 7 years should be provided. 2) [Page 12, Line 50] Similarly, justification or references for the rate of abnormal 30-day CTA following EVAR should be provided. 3) A question of feasibility. Given the estimated sample size requirement of approximately 1,000 patients in each arm of the study (i.e. those with infrequent post-30 day CTA and those with standard annual post-30 day CTA surveillance), and the fact that only one of the likely participating centers indicated that they had adopted the new ESC guidelines for reduced frequency of follow up after a normal post-procedural CTA, do the authors expect that the "reduced-frequency" arm of the study will accrue sufficient patients to meet the enrollment goals of the study? 4) The approach proposed for evaluating long-term survival and freedom from reintervention relies on the use of Danish administrative claims data. Can the authors comment on the risk of loss to follow up for patients who may seek care outside of the Netherlands, and whether this risk might compromise the interpretation of the results of the study?
--

VERSION 1 – AUTHOR RESPONSE

Reviewer(s)' Comments to Author:

Reviewer: 1

Reviewer Name

Janet Powell

Institution and Country

Imperial College London, UK

Please state any competing interests or state 'None declared':

None declared

Please leave your comments for the authors below

Useful study, but may prove dated by time of publication. I have several comments which need to be addressed to improve the protocol (and hopefully the study too).

1 The relevant guideline checklist is STROBE not CONSORT

Response: Thank you, this is correct, the STROBE statement has been replaced for the CONSORT checklist.

2 The outcomes are not clearly listed, particularly the secondary outcomes. For instances costs are mentioned but never listed as a secondary outcome. I suggest giving a complete list of the prespecified outcomes before describing how the data are to be collected.

Response: Good suggestion, a complete overview of the prespecified outcomes has been listed. (page 11-12, lines 221-232:

"Main study endpoint: The number of patients with an intervention and aneurysm related mortality classified for patients with and without yearly imaging surveillance.

Secondary study endpoints:

- Date, type, indication and outcome of all postoperative imaging during follow-up.
- Type I, type II, type III and type IV endoleak, graft or outflow (iliac) occlusion, endograft infection detected by postoperative imaging, if present.
- Date and type of intervention during follow-up, if present.
- Date of aneurysm rupture during follow-up, if present.
- Date of death during follow-up, if present.
- Costs of all EVAR related imaging and outpatient clinic visits.")

The study procedures and how the data is collected is written down after the previous session (page 12, lines 234-264).

3 The covariates to be included in the analyses are not listed and at least a partial list needs to be included in the manuscript.

Response: We have tried to resolve this apparent omission by rewriting the 'statistical methods' section. (page 14-15, lines 301-317: "Multivariable cox regression analysis will be used to determine the freedom of intervention and survival corrected for age, gender, AAA diameter, ASA classification, neck length, neck angulation and type of endograft. The association between postoperative intervention and the following covariates will be investigated with the multivariate Cox-regression analysis:

- age
- gender

- AAA diameter
- ASA classification
- neck length (>15mm)
- neck angulation (>60°)
- type of endograft")
- Initial postoperative CTA

4 You need to consider the major limitation of this study, that it is retrospective and represents the imaging capabilities and expertise of >10 years ago (key issue of first post-operative CT scan). This limitation also is likely to lead to missing data, particularly for covariates, and you need to discuss how missing data are to be handled in the statistics section.

Response: We completely agree with your suggestion and made improvements to the manuscript by adding the requested items to the discussion section. (page 16, lines 339-346: "Our study is subject to limitations due to the nature of administrative data and its retrospective and observational design. As with all studies using administrative data, it allows only the collection of data that was documented in patient medical records. It is also possible that some patients may have transferred to alternative surveillance protocols in different medical centres without our knowledge. The study will assess results in 17 medical centres over 11 years, during which time improvements in endograft and in clinical practice has occurred. Attrition bias due to loss to follow up represents a threat to the internal validity of our cohort study.")

5 How many persons will extract the data and enter it into the data base? There could be a need for at least partial double data extraction to improve the validity of the data.

Response: The validity of the data is sufficient since two persons extract the data and enter it into the data base, we also added it to the manuscript. (page 10, lines 200-204: "At first two researchers will extract data together to standardize data extraction. Next, to further improve the validity of the data two researchers will individually extract data and enter it into the secured data base. Disagreements will be noted and resolved by discussion and if necessary by asking another co-author to act as an arbiter.")

6 Do you need a protocol for extracting data from the radiology and/or duplex reports?

Response: Measuring or verifying these radiology reports does not fall within the scope of this study. We use the measurements made by radiologists in the radiology- and duplex reports, as stated in the manuscript (page 12, lines 251-253: "All imaging outcomes are based on the report compiled by radiologists. These reports will not be re-evaluated by an independent radiologist, since we want to base our outcomes on real life data")

We thank the reviewer for the thorough review of our manuscript and constructive criticism. We hope that the changes made based on these comments have improved our manuscript.

Reviewer: 2

Reviewer Name

Sam Tyagi

Institution and Country

University of Kentucky, USA

Please state any competing interests or state 'None declared':

None declared

Please leave your comments for the authors below

1. Endpoints seem to focus on interventions and aneurysm related mortality. I would suggest categorizing and separating into thrombotic/occlusive/stenotic complications or interventions vs endoleak related vs infection related - there is wide variability in the mortality risk of these categories between and within these categories. If they are all lumped in together, it may confound some real findings. In order to do this, you will need to recalculate your statistical power and sample size.

Response: Thank you for this suggestion. Our primary outcomes are interventions and aneurysm related mortality for patients who had normal initial postoperative CTA and who do adhere to our definition of yearly imaging surveillance of a 6 to 11-year follow-up period, compared to those who do not adhere to our definition. We do categorize all abnormalities found during follow-up imaging and the following complications and interventions based on the Reporting Standards:

<https://www.sciencedirect.com/science/article/pii/S0741521402923864?via%3Dihub>. Our sample size is calculated based on our primary outcomes. The number of patients included in our database allows us to do post-hoc exploratory analyses of differential outcomes according to qualifying event for interventions following EVAR.

2. You protocol lack mention of device specific complications. Some devices may be more prone to others to certain complications (ex. type 3 endoleak), and lumping them all together will confound data, and may mask a real finding.

Response: We agree with the reviewer that some devices are more prone than others and as suggested by the reviewer we added these information to the method section. (page 10, line 188: "Device-specific complications after EVAR will also be examined.")

3. Be aware that your follow up period may still be short. For example the endologix AFX graft with the old material (strata) had a 24% T3EL rate, but the average time to reintervention was 40+ months and up to 70 months.

Response: It would have been very interesting to have longer follow-up, however devices and current practice is changing and therefore might prove dated by time of publication. In the recent study from Väärämäki et al. (EJVES Aug 2019) the overall survival rates was 93%, 61%, 25% and 9% at 1, 5, 10 and 16 years, respectively, stating that most patients have deceased after 10 years.

We thank the reviewer for the thorough review of our manuscript and constructive criticism. We hope that the changes made based on these comments have improved our manuscript.

Reviewer: 3

Reviewer Name

Frederic S. Resnic, MD MSc

Institution and Country

Lahey Hospital and Medical Center, Burlington MA, USA

Please state any competing interests or state 'None declared':

None Declared

Please leave your comments for the authors below

The authors present the very well written protocol for the ODYSSEUS Study of the outcomes following EVAR with different strategies for post-procedural imaging surveillance. The issue being studied is relevant and important to the contemporary treatment of abdominal aortic aneurysms, and

the approach proposed is both a pragmatic and efficient strategy for investigating the research question. I have only a few questions for the authors to consider in the effort to clarify some of the assumptions used in the sample size calculations presented in the protocol.

1) [Page 12, Line 45] More detailed justification or references for the selection of the baseline rate of anticipated intervention-free survival (primary endpoint) of 75% at 7 years should be provided.

Response: We agree with the reviewer pointing out poor referencing and added some references. (page 13, line 270-271, reference 18 and 19)

2) [Page 12, Line 50] Similarly, justification or references for the rate of abnormal 30-day CTA following EVAR should be provided.

Response: We completely agree with your suggestion and made improvements to the manuscript by adding references. (page 13, line 277, reference 20 and 21)

3) A question of feasibility. Given the estimated sample size requirement of approximately 1,000 patients in each arm of the study (i.e. those with infrequent post-30 day CTA and those with standard annual post-30 day CTA surveillance), and the fact that only one of the likely participating centers indicated that they had adopted the new ESC guidelines for reduced frequency of follow up after a normal post-procedural CTA, do the authors expect that the "reduced-frequency" arm of the study will accrue sufficient patients to meet the enrollment goals of the study?

Response: Indeed, only one of the participating centres adapted the new ESVS guideline but with this study we hope to confirm this recommendation. Patients will be divided into three groups A) patients without abnormalities at their first postoperative CTA with yearly imaging surveillance, B) patients without abnormalities at their first postoperative CTA without yearly imaging surveillance and C) patients with abnormalities at their first postoperative CTA.

We do expect enough patients in this arm of the study, we have gathered data from 5 medical centres already and presented our preliminary results at the recent ESVS conference in Hamburg: Total of 679 patients with an infrarenal AAA in the period Jan 2007 – Jan 2012. We included 519 patients without abnormalities at their first CTA, 193 patients (Group A: 37%) were compliant according to our definition and the overwhelming part was non-compliant including 326 patients (Group B: 63%). We excluded 160 patients with abnormalities at their first CTA (Group C).

4) The approach proposed for evaluating long-term survival and freedom from reintervention relies on the use of Danish administrative claims data. Can the authors comment on the risk of loss to follow up for patients who may seek care outside of the Netherlands, and whether this risk might compromise the interpretation of the results of the study?

Response: There are no indications that Dutch people go abroad in large numbers, because health care in the Netherlands is easily accessible for everyone and of good quality.

We thank the reviewer for the positive comments made about our manuscript and the suggested changes. We hope that the answers provided and the changes made are in agreement with the reviewer and have improved our manuscript.

VERSION 2 – REVIEW

REVIEWER	Janet Powell Imperial College London
REVIEW RETURNED	06-Dec-2019

GENERAL COMMENTS	1 As a retrospective study, your objective needs to be rephrased to "The objective of this study is to evaluate whether imaging surveillance frequency might have been safely reduced in a selected group of EVAR patients, for example in patients with an asymptomatic or symptomatic infrarenal AAA who underwent EVAR and who had no abnormalities on the 3 month postoperative CTA." Past behaviour cannot necessarily predict future behaviour, given the advances in imaging, including contrast ultrasound. You also need to specify or standardise the the timing of first post-op CT, partly because small type II endoleaks observed early on can resolve spontaneously. 2 Please provide some information as to how you will handle missing data 3 Please specify whether you can retrieve data on the decision not to correct an endoleak or other problem because of patient frailty. In a long-term study this becomes an issue and might need to be added to your limitations. Also is the decision of a patient to refuse reintervention recorded?
--

REVIEWER	Sam Tyagi University of Kentucky, USA
REVIEW RETURNED	19-Dec-2019

GENERAL COMMENTS	Good revisions to the review of this protocol.
--

REVIEWER	Frederic S. Resnic, MD Lahey Hospital and Medical Center, United States
REVIEW RETURNED	29-Nov-2019

GENERAL COMMENTS	The authors have adequately addressed the concerns raised.
--

VERSION 2 – AUTHOR RESPONSE

Reviewer(s)' Comments to Author:

Reviewer: 1

Reviewer Name
Janet Powell

Institution and Country
Imperial College London

Please state any competing interests or state 'None declared':
None

Please leave your comments for the authors below

1 As a retrospective study, your objective needs to be rephrased to "The objective of this study is to evaluate whether imaging surveillance frequency might have been safely reduced in a selected group of EVAR patients, for example in patients with an asymptomatic or symptomatic infrarenal AAA who underwent EVAR and who had no abnormalities on the 3 month postoperative CTA."

Past behaviour cannot necessarily predict future behaviour, given the advances in imaging, including contrast ultrasound. You also need to specify or standardise the the timing of first post-op CT, partly because small type II endoleaks observed early on can resolve spontaneously.

Response: This is correct, the revision has been made. (page 8 , lines 136-139: "The objective of this study is to evaluate whether imaging surveillance frequency might have been safely reduced in a selected group of EVAR patients, for example in patients with an asymptomatic or symptomatic infrarenal AAA who underwent EVAR and who had no abnormalities on the 3 month postoperative CTA.")

2 Please provide some information as to how you will handle missing data

Response: Following the suggestions of the reviewer, information about handling missing data is added to the 'statistical methods' section. (page 15, lines 311-315: "The proportion of missing data will be displayed. The missing values will be imputed by multiple imputation techniques if this does not exceed 10-15% and conduct a sensitivity analysis to investigate the effect of the missing data on the results of the analysis. If missing data on outcome variables exceeds 15% we perform subgroup analysis.")

3 Please specify whether you can retrieve data on the decision not to correct an endoleak or other problem because of patient frailty. In a long-term study this becomes an issue and might need to be added to your limitations. Also is the decision of a patient to refuse reintervention recorded?

Response: This would indeed be very interesting information, unfortunately this is very hard or almost not to retrieve from patients' medical records. The decision of a patient to refuse reintervention will also not be recorded. This information was added to the limitations-section. (page 17, lines 345-347: "Another limitation is that no information is retrieved from patients' medical records about when not to intervene and what the reason was for this decision.")

Reviewer: 2

Reviewer Name
Sam Tyagi

Institution and Country
University of Kentucky, USA

Please state any competing interests or state 'None declared':
None declared

Please leave your comments for the authors below Good revisions to the review of this protocol.
Response: Thank you.

Reviewer: 3

Reviewer Name
Frederic S. Resnic, MD

Institution and Country
Lahey Hospital and Medical Center, United States

Please state any competing interests or state 'None declared':
None Declared

Please leave your comments for the authors below The authors have adequately addressed the concerns raised.

Response: Thank you.

VERSION 3 - REVIEW

REVIEWER	Janet Powell Imperial College London
REVIEW RETURNED	03-Jan-2020

GENERAL COMMENTS	Thank you for making these latest revisions to improve the description of your interesting study.
---